# Antirheumatic treatment is associated with reduced serum Syndecan-1 in Rheumatoid Arthritis

**Gia Deyab**[1☯*], **Trine Marita Reine**[2,3,4☯], **Tram Thu Vuong**[5‡], **Trond Jenssen**[2,6‡], **Gunnbjørg Hjeltnes**[7‡], **Stefan Agewall**[8,9‡], **Knut Mikkelsen**[10‡], **Øystein Førre**[6‡], **Morten Wang Fagerland**[11☯], **Svein Olav Kolset**[4☯], **Ivana Hollan**[12,13☯]

**1** Department of Laboratory Medicine, Vestre Viken Trust, Drammen, Norway, **2** Section of Nephrology, Department of Transplant Medicine, Oslo University Hospital and University of Oslo, Rikshospitalet, Oslo, Norway, **3** Institute of Cancer Genetics and Informatics, Oslo University Hospital, Oslo, Norway, **4** Department of Nutrition, Institute of Basic Medical Sciences, University of Oslo, Oslo, Norway, **5** Norwegian Veterinary Institiute, Oslo, Norway, **6** Faculty of Health Sciences, Metabolic and Renal Research Group, The Artic University of Norway, Tromsø, Norway, **7** Department of Internal Medicine, Innlandet Hospital Trust, Lillehammer, Norway, **8** Oslo University Hospital Ullevål, Oslo, Norway, **9** Institute of Clinical Sciences, University of Oslo, Oslo, Norway, **10** Department of Rheumatology, Hospital for Rheumatic Diseases, Lillehammer, Norway, **11** Oslo Centre for Biostatistics and Epidemiology, Research Support Services, Oslo University Hospital, Oslo, Norway, **12** Norwegian University of Science and Technology, Gjøvik, Norway, **13** Beitostølen Health and Sport Senter, Beitostølen, Norway

☯ These authors contributed equally to this work.
‡ These authors also contributed equally to this work.
* gyyadeyab@gmail.com

**Data Availability Statement:** All relevant data are within the paper and its Supporting information files.

## Abstract

The endothelial glycocalyx (EG) is essential for proper function of the endothelium and for vascular integrity, but its role in premature atherogenesis in rheumatoid arthritis (RA) has not been studied yet. EG impairment can play a role in pathogenesis of vascular disease, and one of its characteristics is shedding of syndecan-1 from endothelial cells. Syndecan-1 shedding is mediated by matrix metalloproteinase-9 (MMP-9) and counteracted by tissue inhibitor of metalloproteinases (TIMP)-1. Cardiovascular disease risk in RA is reversible by disease modifying antirheumatic drugs (DMARDs), but the exact modes of action are still unclear. Therefore, we examined effects of DMARDs on syndecan-1, MMP-9 and TIMP-1 in RA patients, and searched for associations between these parameters and inflammatory activity. From the observational PSARA study, we examined 39 patients starting with meth-otrexate (MTX) monotherapy (in MTX naïve patients, n = 19) or tumor necrosis factor inhibi-tors (TNFi) in combination with MTX (in MTX non-responders, n = 20) due to active RA. Serum syndecan-1, MMP-9 and TIMP-1 were measured at baseline and after six weeks of treatment. Serum syndecan-1 ($p = 0.008$) and TIMP-1 ($p < 0.001$) levels decreased after six weeks of anti-rheumatic treatment. Levels of MMP-9 also decreased, but the difference was not statistically significant. The improvement in syndecan-1 levels were independent of changes in inflammatory activity. There was no significant difference in changes in synde-can-1 levels from baseline to 6 weeks between the MTX and TNFi groups, however the change was significant within the MTX group. Six weeks of antirheumatic treatment was associated with reduction in serum levels of syndecan-1, which might reflect reduced

**Funding:** The author(s) received no specific funding for this work.

**Competing interests:** The authors have declared that no competing interests exist.

**Abbreviations:** Apo1, Apolipoprotein-A1; CRP, C-reactive protein; CVD, Cardio vascular disease; DAS28, 28-joint Disease Activity Score; ED, endothelial dysfunction; EF, endothelial function; ESR, Erythrocyte sedimentation rate; HbA$_{1c}$, hemoglobin A1c; HDL, high density lipoprotein; ICAM, Intercellular Adhesion Molecule; IL-1β, Interleukin-1β; LDL, low density lipoprotein; MHAQ, medical health assessment questionnaire; MMP-9, Matrix Metalloproteinase-9; MTX, methotrexate; PtGA, Physicians' Global Assessment score of disease activity; RA, rheumatoid arthritis; RDD, Rheumatic Disease Duration; RF, Rheumatoid factor; RHI, Reactive Hyperemia Index; TIMP-1, Tissue-Inhibitor of MMP-1; TNFi, Tumor necrosis factor inhibitors; VAS, Visual Analogue Scale; VCAM, Vascular cell adhesion molecule.

syndecan-1 shedding from EG. Thus, it is possible that EG-preserving properties of DMARDs might contribute to their cardioprotective effects. These effects may be at least partly independent of their anti-inflammatory actions. Our findings do not support the notion that syndecan-1 shedding in RA is mediated mainly by increased MMP-9 or decreased TIMP-9 serum concentration.

## Introduction

Rheumatoid arthritis (RA) is a chronic inflammatory disease that primarily affects joints, but can also affect other organs and tissues including the circulatory system [1]. Cardiovascular disease (CVD), mainly due to atherosclerosis, is the main cause of increased mortality in RA. The pathogenesis of the accelerated atherosclerosis in RA is complex and has not been fully elucidated yet. Among other factors, inflammation appears to play an important role.

RA patients are predisposed to endothelial dysfunction (ED), which is the first and reversible stage of atherogenesis [2]. Therefore, early detection and therapeutic targeting of ED may be of a great clinical importance.

Importantly, antirheumatic treatment has been reported to improve endothelial function (EF) as well as to decrease CVD morbidity and mortality [3,4]. However, the exact underlying mechanisms are still unclear.

Endothelial cells (ECs) delineate the luminal part of vessels, securing vascular integrity. ECs are covered by a protective layer, the endothelial glycocalyx (EG), which is essential for their optimal function. Inflammatory factors known to be upregulated in RA, including tumor necrosis factor (TNF), can trigger disruption of the EG. EG damage leads to increased vascular permeability as well as cholesterol influx and migration of immune cells into the vessel wall, and to reduced ability of the vessel to dilate [5]. Hence, EG impairment can play a substantial role in the pathogenesis of ED and atherosclerosis. Therapies improving EG integrity are therefore called for [2].

Degradation of EG includes shedding of its components, such as syndecans, into the circulation, resulting in their increased serum levels [5]. Syndecans comprise four transmembrane proteoglycans [6,7] residing on the apical and basolateral sides of ECs [8,9]. The ectodomains of syndecans contain long linear glycosaminoglycan (GAG) chains, giving them the unique ability to interact with a wide range of ligands, including cytokines, growth factors and proteases. Consequently, syndecans not only maintain cell homeostasis under normal conditions, but play important roles also under variety of pathologic conditions (e.g., infection, trauma and cancer). Syndecans are involved in multiple aspects of inflammation, from leukocyte recruitment to the resolution of inflammation. Shedding of syndecans is mediated by sheddases, e.g. metalloproteinases (MMPs). In this process, the ectodomains are cleaved from the syndecan core-proteins and released into the circulation, where they can function as autocrine or paracrine effectors [10–12].

Syndecans have been widely studied in the context of inflammation. For example, binding of proinflammatory chemokines to syndecan-1 through its GAG chains allows stabilization of chemokines on the endothelial surface and creates a tethering effect for the chemokine. The subsequent shedding of the chemokine-syndecan-1 complex leads to a stable chemokine gradient, which appears to be crucial for maintaining the directionality of leukocyte migration. Serum syndecan-1 has been proposed as a potential marker of inflammatory activity [13].

Although increased syndecan shedding is well-documented in acute inflammation [13], little is known about its shedding in chronic inflammatory diseases such as RA. Nevertheless, potential roles of syndecans in chronic conditions have been suggested [14]. Syndecan-4 has been reported to have regulatory role in osteoarthritic cartilage, and syndecan-3 to have pro-inflammatory effects in joints of arthritic mice [15–17].

Increased levels of MMP-9, which is involved in syndecan-1 shedding [18], have been reported in RA. MMP-9 as well as its natural inhibitor tissue inhibitors of metalloproteinases (TIMP)-1 might be important not only in regulation of disease activity in RA, but also in pathogenesis of the CVD comorbidity. More information about the process of syndecan-1 shedding in RA is therefore warranted [19–21].

We hypothesized that syndecan-1 levels in RA are related to inflammatory activity, and decrease with antirheumatic treatment. We further hypothesized that syndecan-1 shedding in RA is mediated by MMP-9 and inhibited by TIMP-1.

Therefore, in this study we wanted to examine the effects of antirheumatic treatment on syndecan-1, MMP-9 and TIMP-1, and to search for relationships between these factors and markers of disease activity and CVD risk (such as ED) in RA.

## Material and methods

### Patients

From the Norwegian prospective observational **Ps**oriatic arthritis, **A**nkylosing spondylitis, **R**heumatoid **A**rthritis (PSARA) study, described earlier [22–24], we randomly selected 39 patients starting with either methotrexate monotherapy (MTX group, n = 19) or TNF inhibitor (TNFi) in combination with MTX (TNFi + MTX group, n = 20) due to active RA.

In PSARA study, the treatment regimen was chosen using clinical judgment by rheumatologists not involved in the study, and in accordance with the Norwegian treatment guidelines. All patients starting with MTX were MTX naïve, and all patients starting with TNFi had used MTX earlier without sufficient effect. Briefly, the inclusion criteria comprised age 18–80 years, RA according to the American College of Rheumatology (ACR) 1987 criteria [25], and clinical indication for starting treatment with MTX and/or TNFi. MTX was given in oral doses of 15–25 mg once a week, etanercept as 50 mg subcutaneous injections once a week; adalimumab 40 mg subcutaneous injection every other week; infliximab 3–5 mg/kg intravenous injection at baseline, then following prevailing dosing schedule. Exclusion criteria included clinically significant infections, immunodeficiency, malignancy, pregnancy, breastfeeding, congestive heart failure, uncontrolled diabetes mellitus, recent stroke (within 3 months), systemic glucocorticoid dose > 10 mg/day during the last 2 weeks, or use of TNFi during the last 4 weeks prior to the inclusion. The Regional Ethics Committee for Medical Research approved the study protocol. All the patients were Caucasians and gave written informed consent.

### Clinical and laboratory tests

The patients were examined before and 6 weeks after initiation of DMARD treatment. Data collection included demographic data, medical history, medication, physical findings, self-reported questionnaires and results from laboratory analyses. Venous blood samples were drawn after 8-hours fasting. Hospital routine tests were consecutively performed, while small aliquots of serum were stored at -80˚C for later analyses (including syndecan-1, MMP-9 and TIMP-1 measurements). EF was assessed by the Reactive Hyperemia Index (RHI) measured by a finger plethysmograph (EndoPAT 2000, Itamar) as described previously [26,27], and ED was defined as RHI≤1.67 [28].

## Blood tests

Serum was analyzed in blinded fashion, using ELISA kits for detecting human syndecan-1 (Diaclone, 950.640.192), MMP-9 (R&D, DY911) and TIMP-1 (R&D, DY970), according to the manufacturer's instructions. All standards were within the limits of detection and inter- and intra-assay CV was <10%.

The other circulating parameters were measured as described previously [23,24,29].

## Statistics

For non-normally distributed continuous variables, Mann-Whitney U test was applied for comparison between the treatment groups and Wilcoxon signed rank test for comparisons within a group. For normally distributed variables, independent samples t-test was used for comparison between the two treatment groups and paired t-test for comparison within one group. Categorical data between the treatment groups was compared using the Chi-square test.

We performed linear regression analyses with syndecan-1, TIMP-1, MMP-9, or changes in these variables during the observation period, as the dependent variables, and selected demographic, clinical and laboratory variables as the independent variables. Essential variables (age, gender and treatment group) and variables with $p$-values <0.1 for associations in simple regression analyses were included as independent variables in multiple linear regression analyses. The selection of independent variables for simple linear regression models was guided by clinical judgment.

$P$-values ≤0.05 were considered statistically significant, and all statistical tests were two-sided and performed with SPSS version 24.

# Results

## Patient baseline characteristics

The baseline characteristics listed in Table 1 were similar in both treatment groups, except for longer disease duration and higher occurrence of erosive arthritis in the TNFi+MTX group.

## Effects of antirheumatic treatment on syndecan-1, TIMP-1 and MMP-9

Levels of serum syndecan-1 and TIMP-1 were reduced after 6 weeks of antirheumatic treatment (Table 2). MMP-9 levels also decreased but the change did not reach statistical significance (Table 2).

Syndecan-1 decreased in both treatment groups, and there was no statistically significant difference (baseline-6 weeks) between the two treatment regimens (the decrease in syndecan-1 level reached statistical significance in the MTX, but not in the TNFi group) (Fig 1).

TIMP-1 and MMP-9 decreased in both treatment groups, however the change (baseline-6 weeks) was significant only for TIMP-1 (Fig 1).

## Predictors of baseline levels of syndecan-1, TIMP-1 and MMP-9

We performed simple linear regression analyses with baseline syndecan-1, TIMP -1 and MMP-9, respectively, as the dependent variables, and following independent variables: demographic data [age, gender, education], measures of RA activity and severity [C-reactive protein (CRP), erythrocyte sedimentation rate (ESR), interleukin-6 (IL-6) and TNF, 28-joint Disease Activity Score (DAS28), number of swollen joints, rheumatoid factor, anti-citrullinated antibodies, Medical Heath Assessment Questionnaire (MHAQ)], RA duration, selected markers of CVD risk [smoking, alcohol consumption, exercise, hypertension, BMI, diabetes,

**Table 1. Patient baseline characteristics.**

| | Total n = 39 | MTX n = 19 | TNFi + MTX, n = 20 | *p* for difference between the two treatment groups |
|---|---|---|---|---|
| Age, years | 58.2 (±8,7) | 57.0 (±10) | 59.3 (±8.5) | 0.43 |
| Female, n (%) | 25 (64) | 11 (58) | 14 (70) | 0.44 |
| RA duration, years | 1.00 (0.1–25.0) | 0.1 (0.1–14.0) | 4.5 (0.1–25.0) | <**0.001** |
| **Disease activity** | | | | |
| Erosive arthritis, n (%) | 19 (49) | 4 (21%) | 15 (75%) | <**0.001** |
| DAS28 | 4.96 (±0.97) | 4.90 (±0.90) | 5.03 (±1.06) | 0.68 |
| VAS pain (100-mm scale)) | 57.9 (±22.07) | 59.2 (±21.2) | 56.7 (±23.3) | 0.73 |
| VAS fatigue (100-mm scale) | 50.26 (±25.43) | 43.9 (±28.0) | 56.3 (±21.7) | 0.13 |
| No. of swollen joints | 4.00 (0–28) | 4.00 (2–23) | 5.0 (0–28) | 0.75 |
| No. of tender joint | 7.00 (0–28) | 6.00 (0–24) | 8.00 (2–28) | 0.41 |
| ESR (mm/h) | 22.0 (2–81) | 32.7 (19.7–45.7) | 21.11 (14.0–28.2) | 0.17 |
| CRP (mg/L) | 10.0 (1.0–78.0) | 13.0 (1.0–63.0) | 9.0 (1.0–78.0) | 0.48 |
| PtGA (100-mm scale) | 54.97 (47.92–62.03) | 54.72 (43.92–65.52) | 55.21 (44.98–65.44) | 0.91 |
| MHAQ | 0.67 (0.54–0.80) | 0.63 (0.41–0.84) | 0.71 (0.53–0.89) | 0.37 |
| Anti-CCP, n (%) | 21 (53.8) | 8 (42.1) | 13 (65) | 0.22 |
| RF IgM, n (%) | 24 (61.5) | 11 (57.9) | 13 (65) | 0.71 |
| RF IgA, n (%) | 17 (43.6) | 8 (42.1) | 9 (45) | 0.88 |
| **CVD risk factors, n (%)** | | | | |
| Family history of CVD or death | 20 (51.3) | 10 (50) | 10 (52.6) | 0.26 |
| Any known CVD | 8 (20.5) | 6 (30) | 2 (10.5) | 0.14 |
| Previous cardiac infarction | 4 (10.3) | 3 (15) | 1 (5.3) | 0.33 |
| Angina pectoris | 2 (5.1) | 1 (5) | 1 (5.3) | 0.97 |
| Hyperlipidemia | 7 (17.9) | 4 (20) | 3 (15.8) | 0.74 |
| Hypertension | 14 (35.9) | 9 (45) | 5 (26.3) | 0.24 |
| Current smoker | 6 (15.4) | 2 (10) | 4 (21.1) | 0.35 |
| Diabetes | 3 (7.7) | 3 (15) | 0 (100) | 0.08 |
| RHI | 1.92 (1.42–2.94) | 1.88 (1.42–2.76) | 2.07 (1.50–2.94) | 0.16 |
| **Treatment, n (%)** | | | | |
| Calcium agonist | 5 (12.8) | 3 (15) | 2 (10.5) | 0.69 |
| ACE inhibitors | 5 (12.8) | 2 (10) | 3 (15.8) | 0.60 |
| Beta-blockers | 4 (10.3) | 2 (10) | 2 (10.5) | 0.96 |
| Statins | 9 (23.1) | 6 (30) | 3 (15.8) | 0.31 |
| NSAIDs | 29 (74.4) | 14 (70) | 15 (78.9) | 0.54 |
| Acetyl Salicylic acid | 4 (10.3) | 2 (10) | 2 (10.5) | 0.96 |

*p*<0.05 indicate statistically significant difference between the treatment groups at baseline. Numbers are mean (±SD) or median (range).

TNFi: Anti-tumor necrosis factor, MTX: Methotrexate, RDD: Rheumatic Disease Duration, DAS28: 28-joint Disease Activity Score, VAS: Visual Analog Scale, ESR: Erythrocyte sedimentation rate, CRP: C-reactive protein, PtGA: Patients Global Assessment score of disease activity, MHAQ: Medical Heath Assessment Questionnaire, RF: Rheumatoid Factor; RHI: Reactive Hyperemic Index.

**Table 2. Changes in syndecan -1, MMP-9 and TIMP.**

| Characteristics | Baseline, n = 39 | 6 weeks follow-up, n = 39 | *p* for change |
|---|---|---|---|
| **Syndecan-1 (ng/mL)** | 29.2 (12.6–272.8) | 24.4 (9.9–256.1) | **0.005** |
| **MMP-9 (pg/mL)** | 204 (2.13–819) | 174 (57–1360) | 0.252 |
| **TIMP-1 (pg/mL)** | 355 (151–547) | 257 (120–353) | <**0.001** |

p<0.05 indicate a statistically significant difference between the change from baseline to 6 weeks follow-up. Numbers are median (min-max).

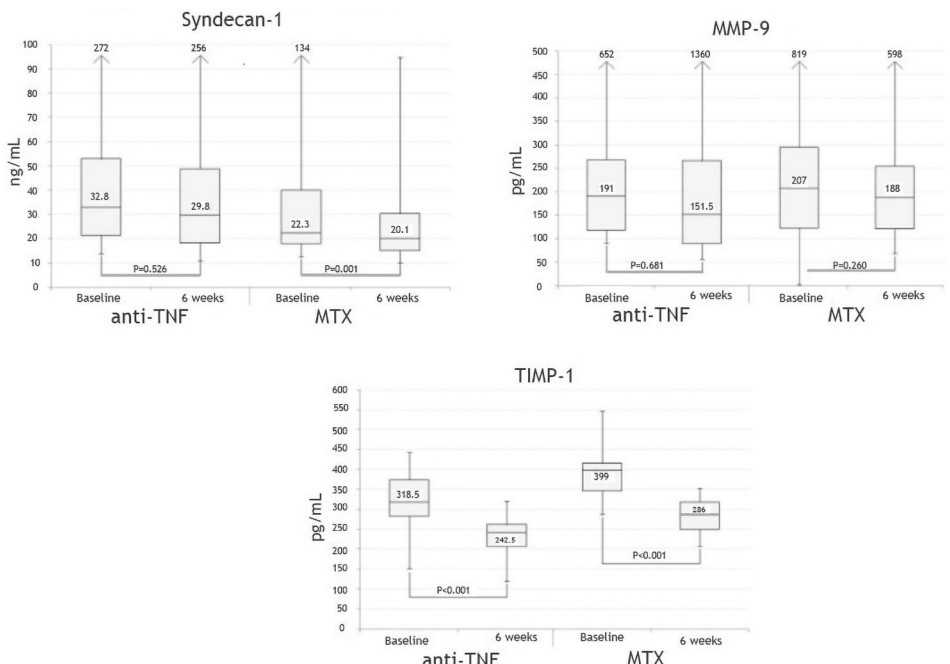

**Fig 1. Effect of treatment.** Boxplot showing levels of syndecan-1, MMP-9 and TIMP-1 at baseline and after 6 weeks of antirheumatic treatment. Midline represents median. Bottom and top of the box represent 25 and 75 percentile and whiskers represent minimum and maximum values.

hyperlipidemia, CVD, family history of CVD, RHI, ED, E-selectin, intercellular adhesion molecule-1 (ICAM-1), vascular cell adhesion molecule-1 (VCAM-1), Apo lipoprotein 1 (Apo B-1), low density lipoprotein cholesterol (LDL-C), high density lipoprotein cholesterol (HDL-C), total cholesterol, serum glucose, hemoglobin A1C (HbA$_1$c)], medications [statins, calcium-blockers, angiotensin-converting-enzyme (ACE) inhibitors, beta-blockers, anti-platelet agents, systemic glucocorticoids, NSAIDs and/or coxibs].

Of these, only CRP was significantly positively related to baseline syndecan-1 level in simple regression analyses (Table 3).

In multiple linear regression analyses adjusted for age, gender and all variables with p<0.1 in simple regression analysis, both CRP and LDL-C were significantly positively related to baseline syndecan-1 (Table 3).

**Table 3. Predictors of baseline syndecan-1.**

| | Unadjusted | | | Adjusted p<0.1 | | |
|---|---|---|---|---|---|---|
| | **Beta** | ***p*** | **95% CI** | **Beta** | ***p*** | **95% CI** |
| Female | 7.845 | 0.628 | -24.692–40.382 | -4.039 | 0.860 | -1.638–1.951 |
| Age | 0.173 | 0.849 | -1.654–2.001 | 0–156 | 0.860 | -1.638–1.951 |
| aTNF+MTX vs MTX | 15.590 | 0.313 | *-15.303–46.483* | | | |
| CRP | 0.881 | **0.043** | 0.028–1.735 | 1.212 | **0.014** | 0.265–2.159 |
| LDL-C | 12.722 | 0.087 | -1.923–27.367 | 18.524 | **0.021** | *2.956–34.091* |
| RHI | 1.379 | 0.069 | -39.293–42.050 | 1.730 | 0.930 | -41.546–38.086 |

TNFi: Anti-tumor necrosis factor; MTX: Methotrexate; CRP: C-reactive protein; LDL-C: Low-density lipoprotein cholesterol; RHI: Reactive hyperaemic index.

In both simple and multiple linear regression analyses adjusted for age and gender, there were no significant associations between neither MMP-9 nor TIMP-1 and any of the examined independent variables at baseline.

### Predictors of change in syndecan-1, TIMP-1 and MMP-9

We performed simple and multiple linear regression analyses examining relationships between changes in syndecan-1, TIMP-1 or MMP-9 as dependent variables and age, gender and change in the following independent variables: CRP, ESR, IL-6, TNF, DAS28, number of swollen joints, rheumatoid factor, anti-citrullinated antibodies, MHAQ, RHI, ED, E-selectin, ICAM-1, VCAM-1, Apo B-1, LDL-C, HDL-C, total cholesterol, serum glucose and HbA$_1$c.

MTX monotherapy was associated with a greater reduction of syndecan-1 than the TNFi regimens (beta = 7.932, $P$ = 0.023, CI (1.164–14.700)). This association remained statistically significant in age and gender adjusted model (beta = 8.111, $P$ = 0.026, CI (1.051–15.172)).

No other significant relationships were observed, neither in simple nor in multiple linear regression analyses.

## Discussion

To our knowledge, this is the first study to demonstrate that antirheumatic treatment is associated with decrease in serum syndecan-1 levels in RA patients. This reduction might be due to decreased syndecan-1 shedding from EG. Thus, the well-known effect of antirheumatic treatment on CVD might be partly due to its protective effect on EG.

It is currently uncertain how DMARDs affect syndecan-1 shedding, and more research in this field is therefore warranted. Understanding of the mode of actions influencing syndecan-1 shedding may contribute to development of novel treatment strategies to protect the glycocalyx and the endothelium.

In theory, syndecan-1 shedding might decrease due to amelioration of inflammatory activity, or due to other glycocalyx preserving effects of DMARDs. Our study indicates that the effect of DMARDs on syndecan-1 shedding might be at least partly independent of their anti-inflammatory effects. In our study, baseline syndecan-1 shedding was related to CRP, but change in syndecan-1 shedding was not related to change in any of the examined inflammatory parameters. One possible explanation could be that although syndecan-1 shedding is related to inflammation, the change in serum syndecan-1 levels induced by DMARDs could be induced by other mechanisms than their anti-inflammatory actions. Nevertheless, also other explanations are possible, including random chance and relatively low sample-size.

Although syndecan-1 has been suggested as a marker of EG, it is important to keep in mind that syndecan-1 is expressed also by other cells, including leukocytes [30]. Therefore, we cannot exclude that the observed decrease in serum syndecan-1 levels does not reflect reduced syndecan-1 shedding (also) from other cells, e.g. immune cells involved in synovitis. Indeed, it is of a great interest to explore not only the effects of DMARDs on integrity of the glycocalyx of ECs, but also of other cell types. Improved insights into the syndecan-1 shedding may help to improve understanding of pathogenesis of RA, with potential consequences for its treatment.

In spite of the role of syndecan-1 in EG, the value of syndecan-1 as biomarker of ED and atherosclerosis is still uncertain. In our study, we did not observe any direct relationship between syndecan-1 and EF. One of the potential explanations could be that syndecan-1 level does not reflect EF in general, or that it does not reflect EF determined by the used method/in the location where we measured it, i.e. in a finger.

Patients in our study had median (range) serum syndecan-1 levels of 29.2 (12.6–272.8) ng/ml. The interpretation of these levels is challenging as information about circulating levels of syndecan-1 in the general population is still limited, and there is a need for establishing reference ranges for clinical use. Nevertheless, the central message from our study is that syndecan-1 levels in active RA appear to be significantly reducible by antirheumatic treatment.

The median syndecan-1 level in our patient group resembles mean syndecan-1 level in patients with coronary artery disease (29.5±4.6 ng/mL), in whom the level increased dramatically (98±9.8 ng/mL) after coronary artery bypass grafting [31,32]. Thus, syndecan-1 levels might be more strongly influenced by acute vascular injury or other tissue damage due to surgery, than by the coronary artery disease [33–35].

To our knowledge, only one study has addressed circulating syndecan-1 levels in RA so far: in a small cohort (18 patients) of relatively young patients (mean age 38,1 ±7.8), syndecan-1 levels were lower than in our study (median 12.8 (8.7–21.5) ng/ml) [36]. These discrepancies might be, among other reasons, due to differences in the patients populations (including age, RA activity, RA severity), and/or differences in laboratory methods.

In the same study, 111 patients with systemic lupus erythematosus had median syndecan-1 levels of 34.2 (20.9–50.0) ng/ml, and the levels were related to disease activity and lupus nephritis.

Previous studies indicate that EG impairment is related to increased risk of CVD. Of the examined traditional CVD risk factors, serum syndecan -1 was only related to LDL-C at baseline, although there was no relationship between changes in these parameters during the observation period.

In an epidemiological study, syndecan-4 shedding was related to myocardial infarction in women [32]. Taken together, as the role of EG destruction in development of CVD is plausible, this topic merits further studies.

The greater syndecan-1 reduction in the MTX group compared to the combined group might indicate that MTX might have a better effect on EG than TNFi, in particular in individuals who respond to its anti-inflammatory activity (as all the patients receiving MTX co-medication in the TNFi group had not responded to preceding MTX monotherapy). Because TNFi is given to patients in whom RA cannot be controlled by MTX, it is also possible that glycocalyx impairment can be more easily reversed at earlier stages of RA and/or in less severe RA than vice versa. As expected, patients in the TNFi group had a longer disease duration than those in the MTX group. Nonetheless, reductions in syndecan-1 levels were seen in both treatment groups (although it was statistically significant only in the MTX group).

Our findings do not support the hypothesis that the improvement in syndecan-1 shedding in RA is mainly due to decrease in MMP-9 and/or increase in TIMP-1 serum levels. We cannot exclude the possibility that the observed reduction in MMP-9 level did not reach statistical significance due to the limited sample size. Further, we cannot exclude a potential role of other inhibitors than TIMP-1 in inhibition of MMP-9. Nonetheless, it seems likely that other MMPs and/or their inhibitors than those examined in our study could play the crucial role in syndecan-1 shedding in RA.

In contrast to our research hypothesis, the levels of TIMP-1 decreased with DMARDs treatment.

Of note, TIMP-1 has multiple functions under physiologic and pathologic conditions, and is involved for example in wound healing and changes in extracellular matrix composition [37]. Thus, future research is needed to determine if TIMP-1 might be a marker of RA activity and/or if improved insights into its functions in RA might help to development of new therapies.

One of the limitations of our study is its observational character. On the other hand, observational studies have also their strengths, such as a greater ability to more accurately reflect real-life. To compensate for the lack of randomization and therefore differences between the two treatment groups, we adjusted for several baseline characteristics using statistical methods.

Although the sample size is relatively low, this explorative study is still important as no other similar studies exist, and as it was able to detect statistically significant differences and associations. Thus, it helps to indicate directions for further research on this topic.

One of the advantages of our study is the comprehensive characterization of the patient population.

In conclusion, six-week antirheumatic treatment was associated with reduction in serum levels of syndecan-1, which might reflect reduced syndecan-1 shedding from glycocalyx. Thus, it is possible that EG-preserving properties of DMARDs might contribute to their cardioprotective effects. These effects may be at least partly independent of their anti-inflammatory actions. Our findings do not support the hypothesis that the improvement in syndecan-1 shedding in RA is mainly due to decrease in MMP-9 and/or increase in TIMP-1 serum levels. Nevertheless, it would be interesting to evaluate in future studies if TIMP-1 could serve as marker of disease activity, and/or if improved insights in the pathophysiologica functions of TIMP-1 could help to development of new therapies in RA.

## Supporting information

**S1 File.**
(ZIP)

## Acknowledgments

We are grateful to the whole PSARA network involved in the establishment and implementation of the PSARA study.

## Author Contributions

**Conceptualization:** Gunnbjørg Hjeltnes, Knut Mikkelsen.

**Formal analysis:** Gia Deyab, Trine Marita Reine, Morten Wang Fagerland, Svein Olav Kolset, Ivana Hollan.

**Investigation:** Gia Deyab, Trine Marita Reine, Gunnbjørg Hjeltnes, Morten Wang Fagerland, Ivana Hollan.

**Methodology:** Trine Marita Reine, Gunnbjørg Hjeltnes, Knut Mikkelsen, Ivana Hollan.

**Project administration:** Trine Marita Reine, Gunnbjørg Hjeltnes, Svein Olav Kolset, Ivana Hollan.

**Resources:** Ivana Hollan.

**Software:** Gia Deyab, Trine Marita Reine.

**Supervision:** Gunnbjørg Hjeltnes, Svein Olav Kolset, Ivana Hollan.

**Validation:** Gia Deyab, Trine Marita Reine, Tram Thu Vuong, Trond Jenssen, Stefan Agewall, Knut Mikkelsen, Øystein Førre, Morten Wang Fagerland, Svein Olav Kolset, Ivana Hollan.

**Writing – original draft:** Gia Deyab, Trine Marita Reine.

**Writing – review & editing:** Gia Deyab, Trine Marita Reine, Tram Thu Vuong, Trond Jenssen, Gunnbjørg Hjeltnes, Stefan Agewall, Knut Mikkelsen, Øystein Førre, Morten Wang Fagerland, Svein Olav Kolset, Ivana Hollan.

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
