## [Decision Letter · Decision Letter 0]

17 May 2021

PONE-D-21-12245

Antirheumatic treatment is associated with reduced serum Syndecan-1 in Rheumatoid Arthritis

PLOS ONE

Dear Dr. Deyab,

Thank you for submitting your manuscript to PLOS ONE. After careful consideration, we feel that it has merit but does not fully meet PLOS ONE’s publication criteria as it currently stands. Therefore, we invite you to submit a revised version of the manuscript that addresses the points raised during the review process.

Both reviewers found some interest in this manuscript, but pointed out several criticisms that require improvement. Of these, lack of considering potential effects of confounding factors on the observed results is critical and requires amendment of the study protocol and additional analysis. I ask the authors to respond to all comments made by the reviewers in the revised version.

We look forward to receiving your revised manuscript.

Kind regards,

Masataka Kuwana, MD, PhD

Academic Editor

PLOS ONE

Journal Requirements:

Reviewers' comments:

Reviewer's Responses to Questions

**Comments to the Author**

1. Is the manuscript technically sound, and do the data support the conclusions?

Reviewer #1: Yes

Reviewer #2: No

2. Has the statistical analysis been performed appropriately and rigorously? 

Reviewer #1: Yes

Reviewer #2: I Don't Know

3. Have the authors made all data underlying the findings in their manuscript fully available?

Reviewer #1: Yes

Reviewer #2: Yes

4. Is the manuscript presented in an intelligible fashion and written in standard English?

Reviewer #1: Yes

Reviewer #2: Yes

5. Review Comments to the Author

Reviewer #1: The present is an interesting study, aiming to evaluate impact of different kind of medications on mediators of atheroslerosis in RA.

Some issues

Methods; reasons for use of TNFi should at least summarized

Methods: the authors speak about multiple regression analysis. Do they means logistic regresssion? cox multivariate analysis (even if the follow up was the same)

Methods: the authors reported B (which in this case probably is OR) and also negative values for CI>should they perform loghartimis transformation to report in a more readiblw way?

Methods/results>should authors perform a multivariate analysis for change in baseline syndecan?

Reviewer #2: Comments for authors

Thank you for your effort. However, this study is quite observational and includes very small number of samples, and seems to lack scientific quality to consider publication in PLOS ONE.

1. Compared to the wide variation of serum syndecan-1 levels at baseline, its change seems quite marginal (although statistically significant). It is difficult for readers to understand whether this minor change leads to cardioprotective effects.

2. There are other factors deeply involved in atherosclerosis of RA such as TNF-α. Were there any correlation with serum syndecan-1 levels and these factors?

3. The quality of the figure is poor and difficult to read.

4. Tables should be inserted between the main manuscript but not at last. Please read the instruction carefully.

6. PLOS authors have the option to publish the peer review history of their article (what does this mean?). If published, this will include your full peer review and any attached files.

Reviewer #1: **Yes: **Fabrizio D'Ascenzo

Reviewer #2: No

---

## [Author Response · Author response to Decision Letter 0]

27 May 2021

Thank you for your comments. We have responded to your comments in a rebuttal letter.

---

## [Decision Letter · Decision Letter 1]

1 Jun 2021

Antirheumatic treatment is associated with reduced serum Syndecan-1 in Rheumatoid Arthritis

PONE-D-21-12245R1

Dear Dr. Deyab,

We’re pleased to inform you that your manuscript has been judged scientifically suitable for publication and will be formally accepted for publication once it meets all outstanding technical requirements.

Kind regards,

Masataka Kuwana, MD, PhD

Academic Editor

PLOS ONE

Additional Editor Comments (optional):

Reviewers' comments:

Reviewer's Responses to Questions

**Comments to the Author**

1. If the authors have adequately addressed your comments raised in a previous round of review and you feel that this manuscript is now acceptable for publication, you may indicate that here to bypass the “Comments to the Author” section, enter your conflict of interest statement in the “Confidential to Editor” section, and submit your "Accept" recommendation.

Reviewer #1: All comments have been addressed

2. Is the manuscript technically sound, and do the data support the conclusions?

Reviewer #1: Yes

3. Has the statistical analysis been performed appropriately and rigorously? 

Reviewer #1: Yes

4. Have the authors made all data underlying the findings in their manuscript fully available?

Reviewer #1: Yes

5. Is the manuscript presented in an intelligible fashion and written in standard English?

Reviewer #1: Yes

6. Review Comments to the Author

Reviewer #1: (No Response)

7. PLOS authors have the option to publish the peer review history of their article (what does this mean?). If published, this will include your full peer review and any attached files.

Reviewer #1: **Yes: **Fabrizio D'Ascenzo

---

## [Editor Report · Acceptance letter]

21 Jun 2021

PONE-D-21-12245R1 

Antirheumatic treatment is associated with reduced serum Syndecan-1 in Rheumatoid Arthritis 

Dear Dr. Deyab:

I'm pleased to inform you that your manuscript has been deemed suitable for publication in PLOS ONE. Congratulations! Your manuscript is now with our production department. 

Kind regards, 

on behalf of

Prof. Masataka Kuwana 

Academic Editor

PLOS ONE